# Acceptability and feasibility of integrating female genital schistosomiasis and sexual and reproductive health interventions in Kenya: A demonstration study

Robinson Karuga[1]*, Millicent Ouma[1], Stephen Mulupi[1], Leora Pillay[2], Caroline Pensotti[3], Victoria Gamba[4], Christine Kalume[2], Delphine Schlosser[2], Isis Umbelino-Walker[5], Ronald Tibiita[2], Florence Wakesho[6], Paul Nawiri[1], Patriciah Jeckonia[1], Thaddeus Owiti[7], Kariuki Njaanake[8], Hannah Ndupha[9], Jackson Muinde[9], Nickson Mugoha[7], Nana Mafimbo[9], Hassan Leli[10], Amos Ndenge[10], Lilian Otiso[1], Julie Jacobson[11]

**1** Department of Research and Strategic Information, LVCT Health, Nairobi, Kenya, **2** Department of Programmes and Partnerships, Frontline AIDS, Brighton, United Kingdom, **3** External Technical Consultant, Bridges to Development, Seattle, Washington, United States of America, **4** Independent Consultant, Nairobi, Kenya, **5** Independent Consultant, Almere, The Netherlands, **6** Division of Vector Borne and Neglected Tropical Diseases, Ministry of Health, Nairobi, Kenya, **7** County Department of Health Services, Homa bay, Kenya, **8** Department of Medical Microbiology and Immunology, University of Nairobi, Nairobi, Kenya, **9** County Department of Health Services, Kwale, Kenya, **10** County Department of Health Services, Kilifi, Kenya, **11** Managing Partner, Bridges to Development, Seattle, Washington, United States of America

\* robinson.karuga@lvcthealth.org

## Abstract

Female genital schistosomiasis (FGS) is a neglected gynaecological condition that is a manifestation of chronic urogenital schistosomiasis. This disease has significant implications for the reproductive health and overall well-being of women and girls, especially in areas with limited access to water, sanitation, and hygiene (WASH). In Kenya, where urogenital schistosomiasis is endemic, the burden of FGS and how to best address it within existing sexual and reproductive health (SRH) services has not been fully explored. This mixed-methods study applied an implementation research approach to assess the feasibility and acceptability of integrating FGS services into routine SRH interventions across public health facilities in three high schistosomiasis endemic counties in Kenya. The intervention included implementing a minimum service package, between December 2023 and December 2024, encompassing health literacy, screening, diagnosis, and treatment. A total of 8,856 women were screened for FGS, with an overall estimated positivity rate of 27.7% (95% CI [26.7, 28.7]). A quantitative survey with a subset of 1,041 clients revealed high acceptability of integration 98.8% (95% CI [98, 99.3]). Integration enabled diagnosis and highlighted a hidden burden of FGS. Qualitative findings revealed significant gaps in knowledge and awareness, stigma-related barriers, and the absence of standardised indicators in the Kenya Health Information System (KHIS), which hampers effective data

**Data availability statement:** All quantitative data are available from the Inter-university Consortium for Political and Social Research (ICPR) Data Repository (ID: ICPSR-237421). Qualitative data may be requested from the LVCT Health Research and Strategic Information Department without restrictions. Please contact enquiries@lvcthealth.org.

**Funding:** This research was funded by the Children's Investment Fund Foundation (CIFF), grant number 2210-08013. The funders had no role in study design, data collection and analysis, decision to publish, or preparation of the manuscript.

**Competing interests:** The authors have declared that no competing interests exist.

**Abbreviations:** CHA, Community Health Assistant; CHP, Community Health Promoter; FGD, Focus Group Discussion; FGS, Female Genital Schistosomiasis; KII, Key Informant Interview; HMIS, Health Information System; MCH, Maternal and Child Health; OPD, Outpatient Department; SSI, Semi-Structured Interview; SRHR, Sexual and Reproductive Health and Rights; WHO, World Health Organization

collection, reporting, and resource planning, including procurement of praziquantel. These findings show the urgent need for health system improvements, including the integration of standardised FGS indicators into the KHIS, to support surveillance, preparedness, and equitable resource distribution, the need for inclusion of FGS within medical training curricula, and for normative clinical guidance on FGS. The evidence supports scaling up FGS-SRH integration and positions MCH clinics and outreach programs as critical entry points.

## Introduction

Female genital schistosomiasis (FGS) is a neglected tropical disease (NTD) with profound implications for health and well-being, particularly for women of reproductive age 15–49 [1,2]. FGS is caused by *Schistosoma haematobium* infection, a species of parasitic flatworm found in freshwater, and is estimated to affect up to 56 million women and girls worldwide [3] who come into contact with contaminated water [4]. FGS affects women and girls living in communities that lack access to adequate water and sanitation services [5,6]. Symptoms of FGS are often non-specific (vaginal discharge, pain, bleeding), and can mimic those of a sexually transmitted infection (STI) [4]. If left untreated, FGS can lead to infertility and other reproductive health issues [7–10], stigma, and social isolation [11–14]. FGS is associated with physiological and immunological changes that can increase the risk of HIV infection and transmission, and speed the progression to AIDS in women who are not on antiretroviral therapy [15,16]. Women with FGS have a three-to four-fold higher risk of contracting HIV infection [15–18]. Additionally, FGS may increase the risk of human papillomavirus (HPV) infection as well as the progression of invasive cervical cancer [6,15,19,20]. Administration of praziquantel, reduces the parasitic burden, alleviates symptoms, and reduces the risk of long-term complications [21]. Despite the scale and severity of FGS, limited awareness of the disease among community members and healthcare workers leads to misdiagnosis as an STI or early-stage cervical cancer, resulting in underreporting, improper treatment, and missed treatment opportunities [14,22].

Strategies to prevent FGS through vertical programmes of vector control [23] and mass drug administration of praziquantel in schools and communities alone cannot address the burden of FGS [24–26]. These programmes typically target school-aged children only, leaving adults untreated. The burden of FGS merits urgent intervention as part of broader sexual and reproductive health (SRH) programming that can reach more women and girls. Despite a growing body of evidence on the relationship between FGS, HIV, HPV, and other reproductive health problems, there is limited information on the programmatic integration of FGS within SRH [27]. Integrating FGS into SRH services is a vital opportunity to offer a comprehensive SRH package for women and girls [11,25,27,28]. Since they do not perceive their SRH needs in isolation, access to accurate information, correct diagnosis, and appropriate care,

including access to water, sanitation, and hygiene (WASH) interventions, is crucial to enable them to achieve optimal health while living stigma-free, healthy, and productive lives.

In Kenya, FGS remains under-researched and underreported, with no prevalence data available despite a high burden of urogenital schistosomiasis among women and girls in some counties, including Homa Bay, Kilifi, and Kwale Counties. The Ministry of Health estimates a *S. haematobium* prevalence of between 2% and 40%, suggesting a significant risk for FGS in these counties [29]. Studies and policy briefs have recommended the adoption of integrated approaches to FGS services to increase the reach and effectiveness of services [1,25]. An implementation research approach was initiated to pilot a minimum service package that guided the integration of FGS into existing SRH interventions in selected sites in Homa Bay, Kilifi, and Kwale Counties [30]. The minimum service package focused on three key areas: health literacy, screening and diagnosis, treatment and care, following the approach of the FGS training competencies [31]. It included a fourth area: social inclusion and equity, which was designed to be included across the three priority components. The intervention involved training healthcare workers from all cadres across the health system about schistosomiasis and FGS and in the delivery of the minimum service package.

This demonstration study aimed to understand how feasible and acceptable it is to integrate FGS services within existing SRH interventions in areas with high endemicity of *S. haematobium* in Kenya. The findings of this project generated evidence to inform scalable and sustainable integration strategies.

## Materials and methods

This was an implementation science study conducted between December 2023 and December 2024. We used a convergent mixed methods approach [32,33] to assess the acceptability and feasibility of integrating FGS and SRH services. The quantitative research component included structured surveys and service delivery data to assess acceptability, while the qualitative research component comprised interviews and focus group discussions with healthcare workers and community members to explore and understand contextual factors pertaining to acceptability and feasibility. We collected data using multiple approaches and from different categories of participants to triangulate the findings and to enhance the depth of interpretation

For this study, we defined acceptability using a multi-faceted construct that is represented by seven components, namely: affective attitude, burden, perceived effectiveness, ethicality, intervention coherence, opportunity costs, and self-efficacy [34] as described in Table 1. We applied the theoretical framework of acceptability to guide the design of data collection tools and analysis of the acceptability of integrating FGS and SRH services.

We also assessed the feasibility of integrating FGS and SRH services. We adopted the definition of feasibility as the practicality and adequacy of the logistics required to integrate FGS and SRH services in resource-limited settings [1,25,35]. We closely monitored activities during preparation for the integration and actual delivery. Our assessment of the feasibility of FGS-SRH integration, looked at the following factors (i) the availability of material resources (equipment, consumables, infrastructure, health registers), (ii) availability and adequacy of health workers (community health workforce, facility based health workers), (iii) comprehensiveness and clarity of information, education and communication (IEC) materials in guiding the health workers on how to integrate FGS and SRH services (training manuals, communication materials) and, (iv) capacity or ability of clients to take up the integrated services. We also explored barriers to and enabling factors for integration.

### Study setting

We purposefully selected Kwale, Kilifi, and Homa Bay Counties for this study due to the high prevalence of urogenital schistosomiasis [29]. We jointly selected sub-counties for the FGS-SRH implementation with the County Health Management Teams (CHMTs) during the inception meetings. This purposeful sampling was based on the MOH granular mapping, which showed high endemicity of *S. haematobium* [29]. The selected sites were required to have existing networks of

**Table 1. Constructs of acceptability based on the theoretical framework of acceptability [33].**

| Acceptability constructs | Definition |
|---|---|
| Ethicality | The extent to which the integration of FGS and SRH services is a good fit with an individual's value system. |
| Affective Attitude | How an individual feels about the integration of FGS and SRH services after taking part. |
| Burden | The perceived amount of effort that is required for clients to access services and for clients and healthcare workers to integrate FGS and SRH services. |
| Opportunity Costs | The extent to which benefits, profits, or values must be given up to engage in the integration of FGS into SRH services. |
| Perceived effectiveness | The extent to which the integration of FGS into SRH services is perceived to be likely to achieve its purpose. |
| Self-efficacy | The participant's confidence that they can perform the behavior(s) required to participate in the integration of FGS and SRH services. |
| Intervention Coherence | The extent to which participants understand the integration of FGS and SRH services and how it works. |

community health units that are linked to primary health facilities and hospitals to assess referral of clients across the different tiers of the health system. A total of nine sites were selected (Homa Bay [two sites], Kilifi [three sites] and Kwale [four sites]). Similar to the rest of the country, there is no data on the burden of FGS in the study counties.

**Kwale County** is located on the south coast of Kenya, covering a total surface area of 8,270.2 square kilometers. It has an estimated population of 866,820. The majority of the population in Kwale County engages in small-scale fishing and subsistence farming. The prevalence of urogenital schistosomiasis ranges between 2.0% and 23.7%. **Kilifi County** is situated on the north coast of Kenya, spanning an area of 12,246 square kilometers and a population of 1,109,735. The prevalence of urogenital schistosomiasis in Kilifi County ranges between 2.0% and 24.4%. **Homa Bay County** is in Western Kenya and has a population of 1,131,950 living within an area of 3,154.7 square kilometres. Lake Victoria is a significant source of livelihood for the residents of Homa Bay County. Residents also use the lake as a source of water for daily household chores and a source of livelihood, such as fishing. The prevalence of urogenital schistosomiasis ranges between 3.0% and 40.4% [29].

## Implementation of the intervention

In 2023, we conducted co-production workshops with the Ministry of Health and County Health Management Teams (CHMTs) in the three study counties to contextualise the minimum service package (MSP), which was developed in an earlier phase of this project [30]. These workshops provided valuable insights about the cultural and socioeconomic considerations to put in place when implementing integrated FGS and SRH interventions based on the MSP across the health system [30]. Table 2 summarises the implementation of the MSP in Kenya.

## Eligibility criteria and recruitment

We applied non-random sampling approaches to enrol study participants between January and December 2024. Before enrolling participants in data collection activities, we determined their eligibility to participate using a screening form. Female clients were eligible to participate in data collection if they were aged above 18 years, with exceptions for emancipated minors (married, sexually active, or had given birth) who were aged above 15 years. Health workers and managers were eligible to participate in the study if they had been in their positions for at least 12 months. We collected data in a language that participants were most comfortable with (either Swahili, Dholuo, or English) after obtaining written informed consent.

**Table 2. Integration of FGS and SRH intervention through implementation of the Minimum Service Package (MSP) in Kenya.**

| MSP Component | Key Activities and Procedures |
|---|---|
| 1. Health Literacy | • Trained 310 CHPs on the signs and symptoms of FGS, myths and misconceptions around FGS, how to use an FGS screening checklist, and refer suspected cases for confirmatory diagnosis through pelvic examination.<br>• Trained 606 facility-based health workers, Community Health Assistants and Public Health Officers on FGS using a cascade approach. We trained 69 healthcare workers as trainers. These trainers then cascaded the training to other healthcare workers through hands-on workshops, mentorship and continuous medical education sessions (CMEs).<br>• Health workers sensitised community members on FGS using community dialogue meetings, outreach activities, facility mini-lectures, and door-to-door sensitisation. |
| 2. Screening & Diagnosis | • Health workers screened women for FGS using a risk checklist that we adapted from the COUNTDOWN training manual [37] (S1 Text) in community and health facility departments such as Maternal & Child Health (MCH), Out Patient Department (OPD), Oncology, among others. The clients who received these integrated services were either referred by CHPs or walked in for other SRH services. CHPs conducted the FGS screening during routine household visits and community outreaches and referred the suspected cases to health facilities for pelvic examination. Clients were referred for pelvic examination if they reported being in contact with stagnant freshwater and had at least one of the symptoms that are indicative of FGS.<br>• Women with suspected FGS after the screening underwent pelvic examination with disposable specula. The diagnosis was based on defined visual FGS indicators.<br>• Modified cervical cancer screening registers to include FGS outcomes since FGS indicators had not been integrated into SRH service delivery registers.<br>• **FGS clinical case definition:** In individuals over the age of 15, who are post-sexual debut and eligible for a pelvic examination:○ **Confirmed Case:** Presence of one or more FGS indicators on pelvic examination. Grainy sandy patches, homogenous yellow patches, rubbery papules, abnormal blood vessels.<br>○ **Probable Case:** One or more self-reported FGS symptoms (Abnormal/copious/odorous vaginal discharge, vulvar/vaginal itching, vulvar mass or ulcer, post-coital bleeding, pain during or after intercourse, vaginal burning sensation/discomfort, bloody discharge, dysuria including incontinence/urge, self-reported genital abnormalities (polyp, cyst)) with no FGS indicators on pelvic examination.<br>• Weekly monitoring and monthly data abstraction; quarterly data quality audits conducted. |
| 3. Treatment & Care | • Diagnosed women received praziquantel (40 mg/kg).<br>• Clients with complications were referred for advanced care.<br>• Clients received a follow-up dose of praziquantel after 8 weeks if symptoms persisted.<br>• Clients whose symptoms did not resolve were advised to return to the health facility for further review. |
| 4.Social Inclusion & Equity | • Health workers provided counselling on FGS prevention and reinfection.<br>• Psychosocial support was offered to those facing stigma or gender-based violence.<br>• Ongoing engagement with county health managers to advocate for budget allocation. |

*Quantitative survey.* Using the sample size formula for estimating proportions in cross-sectional studies [36], we determined that a sample size of 1,050 quantitative survey participants across the three counties would be sufficient to estimate the acceptance of FGS and SRH integration among female clients. In the absence of existing data on acceptance, we assumed a conservative proportion of 50% with a 95% confidence level. We consecutively sampled survey participants after they accessed the integrated FGS-SRH interventions in the nine study health facilities until the determined sample size was achieved. We purposefully invited health care workers who delivered the integrated services to participate in the quantitative acceptability survey.

*Qualitative data collection*. We employed purposive sampling for key informant interviews (KIIs) with health managers and healthcare workers involved in integrating FGS and SRH interventions. Snowball sampling was used to identify community opinion leaders for focus group discussions (FGDs) through collaboration with primary healthcare workers. We consecutively sampled female clients who received integrated services in the health facilities to participate in the semi-structured interviews (SSIs). Additionally, a subset of women who participated in SSIs was purposely sampled for participation in FGDs. The acceptability survey questionnaire, topic guides for interviews, and FGDs are in S2 Text. Table 3 summarises the participants and data collection methods we applied in this study.

**Table 3. Summary of participants and data collection methods by county.**

| Participants | Homa Bay | Kilifi | Kwale | Total |
|---|---|---|---|---|
| | *Number* | *Number* | *Number* | *Number* |
| **Quantitative Survey** | | | | |
| Female Clients | 366 | 350 | 351 | 1067 |
| Healthcare Workers | 15 | 9 | 12 | 36 |
| **FGS Screening and Diagnosis services during community outreach services** | | | | |
| Female Clients | 4,687 | 1,881 | 2,288 | 8,856 |
| **Interviews** | | | | |
| Female Clients | 29 | 34 | 24 | 87 |
| Healthcare Workers | 28 | 18 | 13 | 57 |
| Health Managers | 11 | 8 | 12 | 31 |
| **Focus Group Discussion** | | | | |
| Female Clients | 2 (n = 14) | 2 (n = 16) | 3 (n = 19) | 7 (n = 38) |
| Community Opinion Leaders | 2 (n = 13) | 1 (n = 9) | 2 (n = 27) | 5 (n = 49) |
| Community Health Promoters | 3 (n = 16) | 4 (n = 37) | 3 (n = 24) | 10 (n = 77) |

**Notes**

**Female clients:** These were above 15 years of age who experienced the integrated interventions in the selected primary health facilities and hospitals

**Healthcare Workers:** These included officers in charge of primary health facilities, nurses, clinicians, and supervisors of community health promoters referred to as community health assistants. These health workers were trained on FGS and were responsible for directly providing integrated services to clients.

**Health Managers:** These were professionals who were responsible for managing programs such as neglected tropical diseases, disease surveillance, community health, SRH, and HIV at the national and county levels of the health system.

**Community Opinion Leaders**: These were male and female individuals who were influential in shaping opinions and health-seeking delivery practices in the community. These included community leaders and members of Community Health Committees, which are structures responsible for overseeing the delivery of community health services.

**Community Health Promoters**: They were lay health workers who provided health promotion services and increased health literacy on FGS

## Data collection and analysis

We administered a structured quantitative survey to clients and healthcare workers to assess the different constructs of acceptability toward integrated FGS and SRH interventions: affective attitude, burden, perceived effectiveness, ethicality, intervention coherence, opportunity costs, and self-efficacy. We abstracted data on screening and FGS diagnosis from service delivery registers on a monthly basis. These registers were from the maternity, cervical cancer, HIV, and oncology departments.

All quantitative survey data were collected using an electronic structured questionnaire in Redcap© and uploaded onto a secure server in real-time for cleaning to check for missing variables, inconsistency, and any outliers. These data were then analysed using Stata 17. After the first round of preliminary analysis, we recorded the responses in the five-point Likert scale to a three-point scale as illustrated in the results section. The analysis involved examining statistical measures, including frequency, measures of central tendency (like mean or median), measures of variation (such as standard deviation), and measures of position (like percentiles). The descriptive analysis determined the acceptability of FGS and SRH integration, disaggregated by county. Comparative or inferential statistical analyses were not conducted as they were beyond the scope of this study. The aim was not to establish causal relationships or test hypotheses, but rather to explore the acceptability and feasibility that can inform future, more in-depth analytical studies.

Additionally, qualitative data were collected during interviews and FGDs. We aimed to understand participants' perspectives on the acceptability of the FGS services and their experiences of the integrated interventions. The guides used during the interviews and FGDs explored topics such as the acceptability of integrated FGS services (preferred service points, confidence in integrating these services, modifications required for this integration); experiences and perceptions

of the screening, diagnosis, and treatment for FGS; and the feasibility of integrating these services (the support required to integrate these services, barriers, and facilitators of integrating these services). We also incorporated observations to collect data during the implementation of the MSP, documenting these during weekly study team debriefings.

Qualitative data that were collected during interviews and FGDs were recorded using digital voice recorders. Audio recordings were transcribed verbatim in Microsoft Word and translated from Dholuo or Swahili languages, which are widely spoken at the study sites, into English, where necessary. The translated transcripts were then uploaded into NVivo™ R for analysis. Analysis was conducted inductively. We developed a draft coding framework, initially based on the topics covered in the FGD and SSI topic guides. New codes were added inductively during a data analysis workshop with eight study team members, and the final coding framework was agreed on through consensus. The content of transcripts was systematically coded following a pre-established codebook (S1 Table). The analysis aimed to identify recurring reflections and interpretations.

### Ethical considerations

The Amref Ethics and Scientific Review Committee granted ethical approval for this study (P1510/2023) and the research permit was obtained from the National Council for Science, Technology and Innovation (License No: NACOSTI/P/24/41145). We also received approval to conduct this study from the county-level research departments. We obtained written informed consent from our participants, asking them to participate in all data collection activities. We safeguarded the confidentiality and privacy of study participants by making sure that they could not be linked to any narratives.

## Results

This section presents the characteristics of participants and a concurrent presentation of quantitative and qualitative findings on the acceptability and feasibility of integrating FGS and SRH interventions.

### Participants characteristics

A total of 1,067 female clients participated in the acceptability survey after receiving integrated FGS and SRH interventions (screening and pelvic examination for suspected cases of FGS). After data cleaning, 26 records were removed due to quality reasons such as incompleteness of data, resulting in a final sample of 1,041 women. The mean age of the female clients was 32.8 years (SD = 8.5 years; 95% CI: 32.3 to 33.4). Demographic data on female clients is summarised in Table 4. Additionally, 36 healthcare workers completed the acceptability survey as reported in Table 4. Most healthcare workers were female (n = 21, 58.3%), while (n = 15, 42.7%) were male. In terms of cadres, 58.3% (n = 21) were nurses.

### FGS screening coverage and positivity rates by county and service delivery points

Table 5 summarises FGS screening and positivity rates across three counties and per health service department. Female clients who were screened for FGS were either mobilised during community outreaches, referred by CHPs, or had accessed routine health services in various service departments. FGS positivity is the proportion of female clients who had FGS out of those who underwent pelvic examination.

As illustrated in Table 5, a total of 8,856 clients were screened, with 8,309 referred for pelvic examination in the study health facilities. Among them, 2,301 women were diagnosed with FGS and treated with praziquantel, resulting in an overall estimated positivity rate of 27.7% (95% CI: [26.7, 28.7]). County-specific findings indicate that Kwale had the highest FGS positivity rate at 34.5% (95% CI: [32.6, 36.6]), followed by Homa Bay at 28.5% (95% CI: [27.2, 29.8]), and Kilifi with the lowest rate at 15.4% (95% CI: [13.6, 17.3]). Integration of FGS and SRH services during community outreach identified the highest FGS positivity, followed by the Maternal & Child Health (MCH) and outpatient departments. The Comprehensive Care Clinics had the lowest positivity, likely because their clients reside in urban centers where WASH challenges are generally less prevalent.

**Table 4. Participants' demographic characteristics by county relative to the Acceptability Survey (n = 1,041).**

| Female Clients | Homa Bay | | Kilifi | | Kwale | | Overall | |
|---|---|---|---|---|---|---|---|---|
| | Number | % | Number | % | Number | % | Number | % |
| **Age Categories in years** | | | | | | | | |
| Under 20 years | 7 | 2.1% | 10 | 3.0% | 8 | 2.4% | 25 | 2.5% |
| 20–24 years | 55 | 16.3% | 78 | 23.5% | 41 | 12.4% | 174 | 17.4% |
| 25–29 years | 87 | 25.8% | 71 | 21.4% | 54 | 16.3% | 212 | 21.2% |
| 30–34 years | 63 | 18.7% | 43 | 13.0% | 52 | 15.7% | 158 | 15.8% |
| 35–39 years | 65 | 19.3% | 62 | 18.7% | 62 | 18.7% | 189 | 18.9% |
| 40–44 years | 36 | 10.7% | 33 | 9.9% | 44 | 13.3% | 113 | 11.3% |
| 45–49 years | 24 | 7.1% | 35 | 10.5% | 70 | 21.5% | 129 | 12.9% |
| **Mean Age in years** | Mean | SD | Mean | SD | Mean | SD | Mean | SD |
| | 31.7 | 7.4 | 31.5 | 8.6 | 35.2 | 8.9 | 32.8 | 8.4 |
| **Marital Status** | Number | % | Number | % | Number | % | Number | % |
| Divorced/Separated | 48 | 14.8% | 18 | 5.4% | 30 | 9.1% | 96 | 9.8% |
| Married | 200 | 61.7% | 264 | 79.8% | 228 | 69.1% | 692 | 70.3% |
| Single/Never married | 58 | 17.9% | 43 | 13% | 60 | 18.2% | 161 | 16.4% |
| Widowed | 18 | 5.6% | 6 | 1.8% | 12 | 3.6% | 36 | 3.7% |
| **Currently in school** | Number | % | Number | % | Number | % | Number | % |
| No | 308 | 91.4% | 322 | 96.7% | 283 | 94.3% | 913 | 94.3% |
| Yes | 29 | 8.6% | 11 | 3.3% | 15 | 5.0% | 55 | 5.7% |

**Note:** Missing values were omitted from the analysis

**Table 5. Female clients screened, referred, and diagnosed with FGS by county and health service department.**

| | Screened using the Checklist | Referred for Pelvic Exam | FGS Positive | FGS Positivity | |
|---|---|---|---|---|---|
| | Number | Number | Number | % | 95% CI |
| **By County** | | | | | |
| Homa Bay | 4,687 | 4,575 | 1,304 | 28.5% | [27.2, 29.8] |
| Kilifi | 1,881 | 1,528 | 235 | 15.4% | [13.6, 17.3] |
| Kwale | 2,288 | 2,206 | 762 | 34.5% | [32.6, 36.6] |
| **By Department** | | | | | |
| Oncology | 1,656 | 1,642 | 142 | 8.7% | [73.3, 99] |
| Out-Patient Department | 499 | 87 | 19 | 21.8% | [13.7, 32] |
| Comprehensive Care Clinics | 1,993 | 1,993 | 14 | 0.7% | [0.4, 1.2] |
| Maternal & Child Health (MCH) | 2,190 | 2,189 | 617 | 28.2% | [26.3, 30.1] |
| Maternity | 323 | 323 | 21 | 6.5% | [4.1, 9.8] |
| Community Outreach | 2,195 | 2,075 | 1,488 | 71.7% | [69.7, 73.6] |
| Total | 8,856 | 8,309 | 2,301 | 27.7% | [26.7, 28.7] |

## Acceptability of FGS-SRH integration among female clients and health workers: perspectives across the theoretical framework of acceptability constructs

The acceptability survey results showed that 98.8% (95% CI: [98.0, 99.3]) of female clients across the three counties reported that integrating FGS and SRH was acceptable. Table 6 presents the proportions of acceptability in line with the seven constructs [34]: affective attitude, self-efficacy, burden, effectiveness, opportunity costs, ethicality, and

**Table 6. Female Clients' and Healthcare Workers' acceptability of integrating FGS and SRH using the Theoretical Framework of Acceptability (TFA) constructs [33].**

| TFA Construct | Answer | Female Clients (N = 1,041) | | | Health Workers (N = 36) | | |
|---|---|---|---|---|---|---|---|
| | | *Number* | *%* | *95% CI* | *Number* | *%* | *95% CI* |
| Ethicality | Fair | 1025 | 98.5% | [97.5, 99.1] | 33 | 91.7% | [76.4, 97.4] |
| Affective Attitude | Comfortable | 899 | 86.0% | [84.1, 88.3] | 32 | 88.9% | [73.1, 95.9] |
| Burden | No effort | 1011 | 97.1% | [96.0, 98.0] | 31 | 86.0% | [70.0, 94.2] |
| Opportunity Costs | None | 930 | 89.3% | [87.3, 91.0] | 35 | 97.2% | [81.7, 99.6] |
| Perceived Effectiveness | Agree | 973 | 93.5% | [91.8, 94.8] | 30 | 83.3% | [66.9, 92.5] |
| Self-efficacy | Confident | 1004 | 96.4% | [95.1, 97.4] | 36 | 100% | N/A |
| Intervention Coherence | Agree | 855 | 82.1% | [79.7, 84.3] | 35 | 97.2% | [81.7, 99.6] |
| Overall Acceptability | Acceptable | 1028 | 98.8% | [98.0, 99.3] | 35 | 97.2% | [81.7, 99.6] |

**Note:** Missing data were omitted from this analysis.

intervention coherence. Among the clients, the highest construct was ethicality, 98.5% (95% CI: [97.5, 99.1]). The survey revealed that 97.0% (95% CI: [81.7, 99.6]) of healthcare workers across all sites found the integration of FGS and SRH acceptable. The highest acceptability among health workers was self-efficacy (100%), and the lowest was perceived effectiveness, 83.0%.

Congruent with the quantitative findings, there was unambiguous acceptability of the integrated services among clients, Community Health Promoters (CHPs), and community leaders. Despite the initial anxiety, clients reported that they generally felt safe during the pelvic examination and that they felt relieved that they finally resolved a condition that had been consistently misdiagnosed for years as a sexually transmitted infection (STI). Notably, several CHPs and health workers were diagnosed with FGS and treated. This motivated CHPs and health workers to continue providing health literacy in their communities and referring their clients.

*"I will accept* [the integrated service] *because I came to the hospital, was screened, and found to have FGS. Then I went to Kwale hospital and was treated, and now I am grateful I don't have a problem. I want this organisation to continue; therefore, I am going to tell others to come and get tested"* **(Female CHP Participant, Kwale County)**

We identified two issues that may influence the acceptability of the FGS-SRH integrated intervention, namely gender-related issues and perceived disrespect from health workers. It emerged during FGDs with female CHPs and clients that they preferred being examined by female health workers, during the visual pelvic examination, instead of male health workers. In the same vein, women who were referred for FGS diagnosis reported that it would be important for men in their community to be sensitized about FGS, so that they may allow their spouses to be diagnosed for FGS through visual pelvic FGS examination, as one CHP narrated.

*"The other thing is someone coming to tell you the testing* [referring to pelvic exam] *will be done by a man, I better find a woman […]. So, we are requesting that there should be many women to come and test us because in some of them at home there are men so we were requesting if possible, those men to be put at one place so that they be taught first before us women because it is easy for us women to understand that thing but the men have a problem."* **(Female FGD Participant, Kwale County)**

Some FGD participants reported that the language used by male health workers during the pelvic examination was disrespectful and that it instilled fear and anxiety while undergoing the pelvic examination.

"**R4**: *I was being told, climb the bed, remove your clothes; I was afraid, then he calls another male doctor (sigh) (laughter) that made me fear even more. It was difficult for me to stay in this position (laughter).*

*R3: I didn't know what to do. [The healthcare worker] just forced me to do it. There is a lot of fear, because other people are coming. You need to be told that, when you go there [pelvic examination], you are going to remove your underwear, and you have to sit like this (participant demonstrates). She [female client] should be told clearly, then she will not fear"* (Two CHP FGD Participants, Homa Bay County)

Health professionals responsible for delivering services and managing health programmes accepted the integration of FGS and SRH interventions. After the training on how to diagnose FGS, healthcare workers reported in the SSIs that they could identify FGS and distinguish the lesions from cervical cancer, cervicitis, and STIs through pelvic examination, as reported by one health worker.

"*I can say that, after the first training - the pioneer training that we had - we were able to integrate. Because most of the time, guys have been treating the female genital schistosomiasis as other ailments, like cervicitis. But after getting the training, health workers were able to differentiate between schistosomiasis* [FGS] *and the other cervicitis*" **(Female health worker, Kwale County)**

We also observed that the acceptability of these integrated services was enhanced by the positive treatment outcomes during the study period. FGD participants reported they had been misdiagnosed with other conditions, even with numerous visits to health facilities.

"*Previously, there used to be a lot of quarrels in the family … A man wants to have sex, but the wife says she is having pain, so the husband will say that it is STI because people believed that everything is STI. Like me, I have a neighbour who had that infection. When she went to test, she was FGS positive, she was given medication, and she got well. It is the husband who said with his own mouth, that you can mistake someone for having STI but it is just FGS. So, I think it has helped the community. Because even that lady who was screened has also gone to teach other people in the community*" **(Female CHP Participant, Homa Bay County)**

Healthcare workers indicated that integrating FGS and SRH was cost-efficient, as it eliminated the need for multiple visits to health facilities to test for different conditions.

"*In MCH* [referring to the Maternal & Child Health Department] *we are doing a one-stop shop. When we see the client, we sell all the services. So nowadays there's nothing which needs to be changed. It's a routine. They know we offer all the services*" **(Female health worker, Kwale County)**

Tied to the perceived effectiveness of the integration, healthcare workers also reported that the training on FGS built their confidence in integrating FGS and SRH. They recommended training of other healthcare workers and integrating FGS into the Kenya Nursing curriculum. The data showed that healthcare workers were motivated to integrate FGS and SRH because of the positive outcome of integrating these services.

"*I've seen it work well because there is a client that we have been treating for continuous discharge, lower abdominal pains, but when we integrated and realised what we were told about FGS, we were really on to it* [FGS]. *Now, the clients that were coming back with recurrent discharge. It* [PZQ] *has worked on them, those that we have treated. There's another one* [positive for FGS] *I got last week, and she was very happy*" **(Female health worker, Homa Bay County)**.

**Feasibility of integrating FGS and SRH among healthcare workers**

During interviews with health workers and health managers, we explored the kind of support required for FGS and SRH to be integrated in facilities where the healthcare workers were based, asking respondents for any comments related to service delivery, equipment, human resources, finance, and health information/registers. We also explored barriers to and enabling factors for integration. Healthcare workers and program managers indicated that integrating FGS and SRH is feasible, perceiving integration as a means to enhance the quality of care while requiring a marginal investment in sensitising healthcare workers about FGS.

Healthcare workers and managers emphasised that the successful integration of FGS and SRH interventions will require ongoing investment in training, availability of praziquantel, medical supplies, equipment for diagnosis, and hiring an adequate number of healthcare workers across facilities. As narrated by one health worker, the lack of adequate commodities for diagnosis and medication affects the quality of service provision. In some instances, women of low socio-economic status who could not afford these services stopped seeking care when they were required to pay.

*"...we used to have shortages of commodities and also medication. So the other thing I could say, maybe you can increase the support, maybe if you get like doing the urinalysis, they're being charged some money so once you tell them, you go and do the urinalysis for schistosomiasis so that is the end of it, they don't normally come back….I think the register part is really, really important, especially where, because we lack the reporting tool for the FGS. So I guess if it will be incorporated into the cervical cancer screening register, that will be a great move."* **(Male health worker, Kwale County)**

Healthcare workers reported challenges in documenting FGS in service delivery records. During the demonstration study, they manually added a column to the cervical cancer register to record FGS diagnosis outcomes (positive or negative). Healthcare workers across the study sites preferred integrating an FGS indicator into the cervical cancer register.

*"Yeah, we had a challenge because sometimes you can do screening, and you register from the cervical cancer screening and forget maybe to mention or comment about the FGS because that register is not specifically for that. So, you can say this is not a register for FGS, you have done it but sometimes you forget to fill the results of FGS"* **(Female health worker, Kilifi County)**

County-level managers were supportive of integrating an FGS indicator into the cervical cancer screening register and not adding a new register, as stated by one KII participant: Another health manager expressed frustration with the lack of guidance on how to report FGS from the National Ministry of Health.

*"Nationally, we don't even have tools for reporting* [FGS]. *We have a big problem with tools for reporting. But as I told you earlier, I'll just go and check in KHIS* [Kenya Health Information System] *if FGS has been put in KHIS and if it is there. So maybe the biggest support we can get from the department is… maybe provision... Just… you print a few…"* **(Health manager, Homa Bay County)**

All health managers emphasised that the main barrier to planning and allocating resources for FGS care has been the lack of proper documentation.

Most healthcare workers expressed their confidence in integrating FGS and SRH after receiving the training on screening, diagnosis, and treatment of FGS. CHPs also reported being confident in creating awareness of FGS and referring women for pelvic examination in health facilities.

*"...we have a lot of confidence because we are helping them improve their* [female clients] *lives. We are also helping them to fight myths in their lives. So, this one is a very important thing for them, and we are creating the awareness to know the signs and symptoms of the disease"* **(Female CHP Discussant, Homa Bay County)**

There remains a need for additional mentorship and supportive supervision to develop the confidence of healthcare workers further to make accurate diagnoses, especially for severe cases of FGS, as one healthcare worker reported:

*"At times, it's a bit challenging for some patients or other clients whose cervix has been traumatised; it has undergone certain cellular changes due to trauma or due to STI, recurrent STI. So, I would say I'm at around 95 percent"* **(Female healthcare worker, Kilifi County)**

Most healthcare workers preferred integrating FGS services within Maternal & Child Health Department (MCH) and the SRH service areas, such as family planning. It was also suggested that FGS may be integrated into the cervical cancer screening department. We observed that most health facilities located their cervical cancer screening services in the MCH department.

*"Just like I have said, we integrated it in the MCH and outpatient where we have also females who come there, they are counselled and they are done. We also have the female ward that is in big hospitals like our major hospital, Kwale hospital. Our entry point in Kwale hospital is family planning. It's MCH and family planning clinic, that is where we have integrated outpatient, maternity and female ward. Yes, but for the other rural facilities, because the patients are seen at one place. So, we have totally integrated it into their services so that when they are seeing the females. At least they are able to talk to them about FGS and those who consent are done"* **(Health manager, Kwale County)**

**Perception of workload among healthcare workers**

Most facility-based healthcare workers did not perceive the integration as an extra workload. Only three healthcare workers were concerned that integrating FGS and SRH would increase their workload. They reported that the additional workload was due to healthcare workers feeling duty-bound following training to screen and diagnose FGS via pelvic examination. They also reported that understaffing in their workstations would add to their workload. As represented in this quote, the intervention had created a demand for FGS screening as a result of the higher levels of awareness.

*"The workload has increased because when we started, even those people we had screened before April* [2024], *we were rescreening not for VIA, but now we are looking for FGS. And again, you find the workload has increased because now you have all those women who have gynaecological issues and you see, you treated them twice, thrice* [due to misdiagnosis]. *You'll feel bad if you don't screen them for FGS because maybe that's what is ailing them. So you go an extra mile to screening… So our workload has gone higher and also you find someone coming all away from home. I've come for screening, you cannot tell them I'm busy, go back because they've come from far, they've walked all that distance. Now you can't send them back. So you tell them to wait as they wait, others are waiting. So the workload is becoming hectic, and you get burnout."* **(Female healthcare worker, Kilifi County)**

Some CHPs reported that integrating FGS health literacy added to their workload and some requested additional stipends, in addition to what they receive from the county, to enable them to make multiple visits to their assigned households.

*"...we are requesting a stipend at the end of the month because we are leaving our duties. Like now it's raining, and I cannot weed my farms. You go several times to a household and you don't find anyone. Sometimes they even abuse*

*us, then after some time, you come back and write a referral for them, then they appreciate it after they get help. The workload is big and we are requesting that you give us a small stipend…"* **(Female CHP Participant, Homa Bay County)**

**Factors Influencing the Feasibility of Integrating FGS-SRH**

We observed three notable health system factors that influenced the integration of FGS and SRH services. First, the study sites were also implementing vertical cervical cancer screening programs for women living with HIV. In these vertical programs, essential commodities such as specula were procured exclusively for women enrolled in the HIV programs and could not be used for pelvic examinations on other women, even when county-procured supplies were depleted. This resulted in healthcare workers being frustrated. Secondly, we observed that human resource management was critical to integration. Disruptions occurred when trained healthcare workers went on leave or were transferred to other facilities without complete handover processes, impacting the continuity of integrated services. This frustrated CHPs who had referred women for diagnosis after screening them at the household level. Third, we observed that it was a challenge for CHPs to track and document the uptake of referrals for FGS diagnosis. Some women who were referred for FGS diagnosis sought services in private and other public health facilities. This may have presented an impression that women did not take up referrals by CHPs.

County health managers and healthcare providers shared recommendations on how to address bottlenecks that may impede the integration of FGS and SRH. Table 7 presents key recommendations to address barriers hindering the integration of FGS, highlighting practical insights and challenges related to funding, training, staffing, equipment, medication, documentation, and curriculum inclusion.

## Discussion

This mixed-methods study demonstrates that integrating FGS services with routine SRH interventions using the Minimum Service Package (MSP) [30] is feasible and acceptable within public health systems in counties with a high endemicity of

**Table 7. Recommendations and Supporting Narratives to Address Barriers to the Integration of Female Genital Schistosomiasis (FGS) into SRH Interventions.**

| Recommendation to address barriers for integration | Narrative |
| --- | --- |
| Documentation of FGS in service delivery registers | *"On M&E, we have a challenge in documentation…we don't have a designated data collection tool for FGS reporting […] even in the cases that are diagnosed, we are not able to gather enough documentary evidence […] the routine data collection tools do not have a space to report this. So, gathering that data has been a challenge"* **(Health manager, Homa Bay County)** |
| Advocate for adequate financial resources from county governments for FGS-SRH integration | *"…praziquantel* [PZQ] *is very expensive and it is not routinely supplied by the Kenya Medical Supply Agency. So, it requires additional funding from the county government to procure it […] So, advocating for the county government to buy PZQ is a very tall order[…] we are surviving on donations but we hope that with time the county government will see the sense in procuring this"*. **(Health Manager Homa Bay County)** *Trainings of course to the staff and then also staffing as need might require and also some equipment for diagnosis[…] Since there will be diagnosis, […] we expect the number of patients suffering from the condition to increase so they will need treatment"*. **(Health Manager Kwale County)** |
| Increased staffing to meet demand | *"Okay, the barriers could be the* [human] *resources […] because already the staff, the healthcare workers are carrying out other duties. So with increased sensitization, we end up with more clients, there is that increased burden to the clinics and also the healthcare workers […] that could be one of the immediate bottlenecks that could be associated with the integration as of now.."* **(Health Manager, Homa Bay County)** |

urogenital schistosomiasis in Kenya. We demonstrated that integration enables screening, diagnosis, and treatment. We will discuss the main findings in three themes. Firstly, we will examine the overall feasibility and acceptability of integrating FGS screening into SRH interventions using the MSP, highlighting how this framework supports healthcare workers in identifying and managing FGS cases and the need for further research on social inclusion and equity strategies. Secondly, we will explore health systems factors affecting integration, including the absence of standardised indicators in the Kenya health information systems (KHIS) and priority entry points for integrated FGS-SRH service delivery. Lastly, we will discuss policy and health system strengthening measures needed for sustainable integration.

### Feasibility and Acceptability of FGS Integration Using the Minimum Service Package (MSP)

Our study demonstrated that the MSP [30] is a suitable framework for integrating FGS screening, diagnosis and treatment into SRH interventions in Kenya, addressing key gaps in health literacy, screening, diagnosis, and treatment. Using the Theoretical Framework of Acceptability (TFA) [34], our quantitative findings indicate that female clients, healthcare workers, and managers found the integration to be acceptable and feasible. Similar studies on integration conducted in the Ivory Coast [37], Nigeria [24], South Africa [23], Ghana and Madagascar [28] have emphasised health literacy, diagnosis, and treatment, with training interventions being widely accepted. Qualitative data revealed concerns about the ethicality construct [34] among female clients, particularly regarding stigma, privacy and the need to seek consent from the male partner. Experiences of disrespectful treatment by healthcare workers conducting pelvic examinations may hinder the uptake of FGS diagnosis and treatment. This challenge is consistent in SRH programs where pelvic examinations are performed and needs to be addressed [38,39]. Healthcare workers, while confident in conducting pelvic examinations, expressed concerns about their self-efficacy, particularly regarding challenges in resource constraints [28,37]. Healthcare managers highlighted the need for financial support, improved infrastructure and resources and wider capacity building of healthcare workers. Consistent with several other FGS studies, persistent constraints such as limited diagnostic commodities and the availability of praziquantel remain a challenge [22,23,28,37].

Our results highlight the urgent need to strengthen the social inclusion and equity aspects of the MSP implementation in the Kenyan context, ensuring integration efforts extend beyond biomedical approaches to a more patient-centred and human rights-based model [11,25,30]. Similar to other studies, our findings show that training healthcare workers can improve client-centred inclusive approaches. However, challenges persist in reducing the risk of exposure to contaminated water, ensuring equal access to praziquantel, and in accessing services to address gender-based violence [12,14]. The MSP framework [30] provides a structured approach to addressing these issues, but additional efforts are required in contextualization to ensure culturally sensitive care, respectful provider-patient interactions, and the integration of community-based approaches to improve awareness and service accessibility [38,39]. Strengthening these social dimensions is critical for reducing health inequities and fostering trust in FGS-related care, ultimately supporting the long-term success of integrated service delivery [11,40].

### Addressing health system challenges

Our study highlights critical health system challenges. The absence of standardised indicators in Kenya health information systems (KHIS), hinders data collection and reporting. FGS data is needed to inform resource allocation, as is also seen in similar studies [23,24,28,37]. For instance, decision makers need data on FGS to inform procurement of praziquantel and procurement and distribution of staff and equipment. Investment in standardised reporting indicators for FGS and inclusion of FGS indicators in the HMIS is imperative to break the cycle of neglect, track the FGS burden, inform preparedness, and enhance resource allocation to improve service delivery. While cervical cancer screening programs and HIV clinics have traditionally been key points of entry for FGS diagnosis [25], in our qualitative study and the FGS screening coverage reports, the healthcare workers and managers, and FGS positivity data suggest that Maternal and Child Health (MCH) and community outreach services should also be prioritised.

**Strengthening policy and health systems for sustainable integration**

Based on our findings, we recommend the following actions to strengthen policy and health systems for the sustainable integration of FGS into SRH interventions in Kenya:

1. **Promote broader service integration:** While integration of FGS within SRH interventions demonstrated high levels of acceptability and feasibility, our qualitative findings indicate that vertical health programs, such as those for cervical cancer screening, restrict the flexibility of healthcare workers to provide comprehensive care. To mitigate this challenge, policies should support cross-program integration, allowing resources and training to be utilised across different SRH services.

2. **Priority entry points for service integration:** Prioritise community outreaches and MCH and outpatient entry points for FGS service integration. These service points demonstrated potential for integrating screening, diagnosis, and treatment of FGS alongside routine SRH services. Additionally, integrating FGS services in youth friendly SRH clinics can provide health literacy, screening, diagnosis and treatment services to sexually active youth and adolescents who are in contact with stagnant water in endemic regions.

3. **Include FGS in medical training curricula and normative guidelines.** Ensure FGS is included in healthcare workers and health manager training, including continuous medical education and supportive supervision. This should include nursing, midwifery, and community healthcare worker pre-service training. The integration of FGS screening, diagnosis and treatment of FGS in normative SRH guidelines will help in standardising the management of FGS and capacity building.

4. **Incorporate FGS into Health Management Information Systems (HMIS):** To improve data collection, disease surveillance, and inform resource allocation, FGS should be included as a standardised indicator in HMIS. This will enable policymakers and health managers to track FGS trends, make evidence-based decisions, and allocate appropriate resources to address this neglected condition.

5. **Ensure Sustainable Procurement of Commodities for Diagnosis and Treatment:** Sustainable financing for procurement and distribution is needed to ensure continuous availability of praziquantel, consumables, and necessary equipment. This will prevent stockouts and ensure that integrated services remain functional.

6. **Scale-Up FGS-SRH Integration:** Our study highlights that using the MSP framework for integrating FGS screening and diagnosis within SRH settings is both acceptable and feasible. To build on this evidence, sub-national health authorities should allocate adequate resources, including continuous capacity building for healthcare workers and the provision of essential commodities such as speculums and praziquantel. Expanding integration efforts will ensure that vulnerable and neglected populations are reached effectively.

## Limitations

Our study has some limitations. (i) We did not conduct comparative or inferential statistical analyses, as they were beyond the scope of this study. While this limited the depth of quantitative analysis, the integration of qualitative data offers valuable contextual insights and contributes to triangulation. (ii) There may have been social desirability bias, where healthcare workers and clients responded to our survey questions and SSIs in a way that made us view them favourably. We made efforts to mitigate this bias by explaining the objectives of our study and assuring the participants of confidentiality and that the data collection was not an appraisal of healthcare worker knowledge and skills. We only administered the survey to assess the acceptability of this integration to clients who accessed services in health facilities. We may have missed the perspectives of clients who were diagnosed with FGS in community settings during the outreach. (iii) This being a demonstration study, we did not select the community settings with the highest prevalence of urogenital

schistosomiasis since there is heterogeneity in the prevalence of urogenital schistosomiasis within these counties. The FGS positivity may be higher in the study counties. We quantified this uncertainty by calculating the 95% confidence interval of the FGS positivity (iv). Finally, we conducted this study in public health facilities only. It is possible that women who were referred for pelvic examination may have taken up the services in privately owned health facilities where we had not trained healthcare workers on FGS. This may have underestimated the FGS positivity. This limitation may be overcome by scaling up the sensitisation of FGS to all health workers on how to screen, diagnose and treat FGS, including those in privately owned facilities.

## Conclusions

The integration of FGS and SRH using the MSP is highly acceptable and feasible. This demonstration study shows that the screening, diagnosis, and treatment of FGS in SRH settings are both efficient and scalable in regions with high prevalence of urogenital schistosomiasis. To ensure the sustainability of this integration, it is essential to incorporate FGS as an indicator in the health management information system to facilitate reporting, commodity supply, and allocation of resources for training health workers, diagnosing, and treating FGS.

## Supporting information

**S1 Text: FGS Risk Screening Checklist.**
(DOCX)

**S2 Text: Data Collection Tools.**
(DOCX)

**S1 Table: Code Book.**
(DOCX)

## Acknowledgments

We sincerely thank Anne Wanjiro, Anthony Mwaniki, Darleen Agingu, Filia Gatwiri, Gerald Alambo, Kiti Mwangome, Mercy Kimotho, Millicent Oswe, Oloo Thomas, Sakibu Lyaga, Teddy Kotho and Yassin Mwakaribu for their dedication and invaluable contributions to the implementation study, data collection, and fieldwork. We thank all study participants for their time, trust, and willingness to share their perceptions and experiences in this study. We thank the leadership in the Ministry of Health's NTD Division and the County Health Management Teams for their insights and support throughout the study. In a special way, we thank Wykliffe Omondi, Sharon Chebet, Dr. Hajara El Busaidi, Dr. Gordon Okomo, Dr Michael Audo, Dr. Patrick Oyaro, Meshack Mwandeje, Stephen Wagude and Justus Ocholla for their support in making this study successful. We also thank the leadership at LVCT Health, Frontline AIDS and Bridges to Development for their support in this study. Finally, we extend our gratitude to the Children's Investment Fund Foundation (CIFF) for their generous funding support and commitment to advancing research on the integration of FGS and SRH.

## Financial disclosure

This research was funded by the Children's Investment Fund Foundation (CIFF), grant number 2210–08013. The funders had no role in study design, data collection and analysis, decision to publish, or preparation of the manuscript.

## Author contributions

**Conceptualization:** Robinson Karuga, Leora Pillay, Caroline Pensotti, Christine Kalume, Ronald Tibiita, Patriciah Jeckonia, Lilian Otiso.

**Formal analysis:** Robinson Karuga, Millicent Ouma.

**Funding acquisition:** Robinson Karuga, Christine Kalume, Patriciah Jeckonia, Lilian Otiso.

**Investigation:** Robinson Karuga, Millicent Ouma, Stephen Mulupi, Leora Pillay, Caroline Pensotti, Victoria Gamba, Christine Kalume, Delphine Schlosser, Florence Wakesho, Paul Nawiri, Thaddeus Owiti, Hannah Ndupha, Jackson Muinde, Nickson Mugoha, Hassan Leli, Amos Ndenge, Julie Jacobson.

**Methodology:** Robinson Karuga, Stephen Mulupi, Caroline Pensotti, Christine Kalume, Kariuki Njaanake.

**Project administration:** Robinson Karuga, Millicent Ouma, Stephen Mulupi, Christine Kalume, Delphine Schlosser, Ronald Tibiita, Lilian Otiso, Julie Jacobson.

**Supervision:** Robinson Karuga, Millicent Ouma, Caroline Pensotti, Victoria Gamba, Christine Kalume, Ronald Tibiita, Thaddeus Owiti, Hannah Ndupha, Jackson Muinde, Nickson Mugoha, Nana Mafimbo, Hassan Leli, Amos Ndenge, Lilian Otiso.

**Validation:** Robinson Karuga, Millicent Ouma, Leora Pillay, Caroline Pensotti, Victoria Gamba, Christine Kalume, Isis Umbelino-Walker, Ronald Tibiita, Florence Wakesho, Kariuki Njaanake, Hannah Ndupha, Jackson Muinde, Nana Mafimbo, Lilian Otiso, Julie Jacobson.

**Visualization:** Robinson Karuga, Isis Umbelino-Walker, Paul Nawiri.

**Writing – original draft:** Robinson Karuga, Millicent Ouma, Stephen Mulupi, Leora Pillay, Caroline Pensotti, Victoria Gamba, Christine Kalume, Delphine Schlosser, Isis Umbelino-Walker, Ronald Tibiita, Florence Wakesho, Paul Nawiri, Patriciah Jeckonia, Thaddeus Owiti, Kariuki Njaanake, Hannah Ndupha, Jackson Muinde, Nickson Mugoha, Nana Mafimbo, Hassan Leli, Amos Ndenge, Lilian Otiso, Julie Jacobson.

**Writing – review & editing:** Robinson Karuga, Millicent Ouma, Stephen Mulupi, Leora Pillay, Caroline Pensotti, Victoria Gamba, Christine Kalume, Delphine Schlosser, Isis Umbelino-Walker, Ronald Tibiita, Florence Wakesho, Paul Nawiri, Patriciah Jeckonia, Thaddeus Owiti, Kariuki Njaanake, Hannah Ndupha, Nickson Mugoha, Nana Mafimbo, Hassan Leli, Amos Ndenge, Lilian Otiso.

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
