## [Decision Letter · Decision Letter 0]

1 Aug 2025

PGPH-D-25-01741

Acceptability and feasibility of integrating female genital schistosomiasis and sexual and reproductive health interventions in Kenya: a demonstration study

Dear Dr. Karuga,

Thank you for submitting your manuscript to PLOS Global Public Health. After careful consideration, we feel that it has merit but does not fully meet PLOS Global Public Health’s publication criteria as it currently stands. Therefore, we invite you to submit a revised version of the manuscript that addresses the points raised during the review process.

Please see the attached document and comments below for specific recommendations and suggestions. 

We look forward to receiving your revised manuscript.

Kind regards,

Claire J Standley

Academic Editor

Journal Requirements:

1. We ask that a manuscript source file is provided at Revision. Please upload your manuscript file as a .doc, .docx, .rtf or .tex.

2. Please provide separate figure files in .tif or .eps format.

3. We have amended your Competing Interest statement to comply with journal style. We kindly ask that you double check the statement and let us know if anything is incorrect.

4. Thank you for uploading your study's underlying data set. Unfortunately, the repository you have noted in your Data Availability statement does not qualify as an acceptable data repository according to PLOS's standards.

Additional Editor Comments (if provided):

In addition to the peer review feedback (see attached), please find some additional minor comments and suggestions for consideration:

- Line 124: Change “9” to “nine”. All numbers under 10 should be spelled out in full.

- Lines 121-131: Consider removing these lines – while the additional context for why Sh transmission is high in these counties is interesting, it’s not directly relevant to the paper (since it has already been established that surveys have shown high prevalence of urogenital schistosomiasis in these counties). Please keep the last sentence, related to the lack of data on burden of FGS in these counties.

- Line 252: Please be consistent with use of decimals/number of significant figures listed – i.e. suggest changing 97% to 97.0% (if indeed that was the original result).

- Line 272: The “they” in this sentence is ambiguous – I believe it refers to the female patients (i.e. the female patients need permission from male partners to undergo a pelvic exam) but it could be misunderstood as referring to the men. Please adjust.

- Line 335: Should this be a sub-heading, i.e. it falls under the broader umbrella of “Feasibility of integrating FGS and SRH among healthcare workers”?

- Lines 541-543: This seems to contradict the Methods, where it is stated that the three counties were selected due to high prevalence of urogenital schistosomiasis. Or is the point here that within each county, there may be heterogeneity? Please clarify.

Reviewers' comments:

Reviewer's Responses to Questions

**Comments to the Author**

1. Does this manuscript meet PLOS Global Public Health’s publication criteria?

Reviewer #1: Yes

2. Has the statistical analysis been performed appropriately and rigorously?

Reviewer #1: Yes

3. Have the authors made all data underlying the findings in their manuscript fully available (please refer to the Data Availability Statement at the start of the manuscript PDF file)?

Reviewer #1: Yes

4. Is the manuscript presented in an intelligible fashion and written in standard English?

Reviewer #1: Yes

Reviewer #1: This is a well-written manuscript. Limitations were clearly outlined including limitations in conducting comparative or inferential statistical analyses. This study comes as a useful tool for assisting programme managers in integrating urogenital schistosomiasis into exiting SRH in endemic countries.

The comments are intended to support advocacy efforts for the prioritization of urogenital schistosomiasis, a condition that remains significantly neglected in current health agendas.

**Do you want your identity to be public for this peer review?** For information about this choice, including consent withdrawal, please see our Privacy Policy

Reviewer #1: **Yes: ** Takalani Girly Nemungadi

---

## [Editor Report · Decision Letter 1]

29 Aug 2025

Acceptability and feasibility of integrating female genital schistosomiasis and sexual and reproductive health interventions in Kenya: a demonstration study

PGPH-D-25-01741R1

Dear Dr. Karuga,

We are pleased to inform you that your manuscript 'Acceptability and feasibility of integrating female genital schistosomiasis and sexual and reproductive health interventions in Kenya: a demonstration study' has been provisionally accepted for publication in PLOS Global Public Health.

Best regards,

Claire J Standley

Academic Editor

Thank you for the careful consideration of the peer review feedback!